# Phenotype inference in an *Escherichia coli* strain panel

**Marco Galardini[1], Alexandra Koumoutsi[2], Lucia Herrera-Dominguez[2], Juan Antonio Cordero Varela[1], Anja Telzerow[2], Omar Wagih[1], Morgane Wartel[2], Olivier Clermont[3,4], Erick Denamur[3,4,5], Athanasios Typas[2]\*, Pedro Beltrao[1]\***

[1]European Molecular Biology Laboratory, European Bioinformatics Institute (EMBL-EBI), Hinxton, United Kingdom; [2]Genome Biology Unit, European Molecular Biology Laboratory (EMBL), Heidelberg, Germany; [3]INSERM, IAME, UMR1137, Paris, France; [4]Université Paris Diderot, Paris, France; [5]APHP, Hôpitaux Universitaires Paris Nord Val-de-Seine, Paris, France

**Abstract** Understanding how genetic variation contributes to phenotypic differences is a fundamental question in biology. Combining high-throughput gene function assays with mechanistic models of the impact of genetic variants is a promising alternative to genome-wide association studies. Here we have assembled a large panel of 696 *Escherichia coli* strains, which we have genotyped and measured their phenotypic profile across 214 growth conditions. We integrated variant effect predictors to derive gene-level probabilities of loss of function for every gene across all strains. Finally, we combined these probabilities with information on conditional gene essentiality in the reference K-12 strain to compute the growth defects of each strain. Not only could we reliably predict these defects in up to 38% of tested conditions, but we could also directly identify the causal variants that were validated through complementation assays. Our work demonstrates the power of forward predictive models and the possibility of precision genetic interventions.

DOI: https://doi.org/10.7554/eLife.31035.001

**\*For correspondence:**
typas@embl.de (AT);
pbeltrao@ebi.ac.uk (PB)

**Competing interests:** The authors declare that no competing interests exist.

## Introduction

Understanding the genetic and molecular basis of phenotypic differences among individuals is a long-standing problem in biology. Genetic variants responsible for observed phenotypes are commonly discovered through statistical approaches, collectively termed Genome-Wide Association Studies (GWAS, *Bush and Moore, 2012*), which have dominated research in this field for the past decade. While such approaches are powerful in elucidating trait heritability and disease associations (*Yang et al., 2010*; *Welter et al., 2014*), they often fall short in pinpointing causal variants, either due to lack of power to resolve variants in linkage disequilibrium (*Edwards et al., 2013*) or for the dispersion of weak association signals across a large number of regions across the genome (*Boyle et al., 2017*). Furthermore, by definition GWAS studies are unable to assess the impact of previously unseen or rare variants, which can often have a large effect on phenotype (*Bodmer and Bonilla, 2008*). Therefore, the development of mechanistic models that address the impact of genetic variation on the phenotype can bypass the current bottlenecks of GWAS studies (*Lehner, 2013*).

In principle, phenotypes could be inferred from the genome sequence of an individual by combining molecular variant effect predictions with prior knowledge on the contribution of individual genes to a phenotype of interest. Such knowledge is now readily available by chemical genetics approaches, in which genome-wide knock-out (KO) libraries of different organisms are profiled across multiple growth conditions (*Kamath et al., 2003*; *Dietzl et al., 2007*; *Hillenmeyer et al.,*

*2008*; *Nichols et al., 2011*; *Price et al., 2016*). One output of such screens are genes whose function is required for growth in a given condition (also termed 'conditionally essential genes'). Variants negatively affecting the function of those genes are likely to be associated with individuals displaying a significant growth defect in that same condition. This approach would then effectively be a way to prioritize variants with respect to their impact on the growth on a particular condition. The impact of variants on gene function can be reliably inferred using different computational approaches (*Thusberg et al., 2011*; *Kulshreshtha et al., 2016*), which can offer mechanistic insights into how such variants impact on a protein's structure and function. As rare or previously unseen variants can also be used by this approach, there lies the possibility to deliver predictions of phenotypes at the level of the individual, with no prior model training. Combining variant effector predictors with gene conditional essentiality has been previously tested with some success in the budding yeast *Saccharomyces cerevisiae*, but only on a limited number of individuals and conditions (15 and 20, respectively, *Jelier et al., 2011*). Given the continuous increase in genome-wide gene functional studies, there is an opportunity to apply such an approach more broadly.

In contrast to eukaryotes, diversity within the same bacterial species can result in two individuals differing by as much as half of their genomic content. *Escherichia coli* is not only one of the most studied organisms to date (*Blount, 2015*), but also one of the most genetically diverse bacterial species (*Lukjancenko et al., 2010*; *Tenaillon et al., 2010*). Individuals of this species (strains), exhibit a diverse range of genetic diversity, from highly homologous regions to large differences in gene content, collectively termed 'pan-genome' (*Medini et al., 2005*). Phenotypic variability is therefore likely to arise from a combination of single nucleotide variants (SNVs) and changes in gene content. Since gene conditional essentiality has been heavily profiled for the reference *E. coli* strain (K-12, *Nichols et al., 2011*; *Price et al., 2016*), here we set out to systematically test the applicability of such genotype-to-phenotype predictive models for this species. We reasoned that it would also test the limits of the underlying assumption of this model, which is that the effect of the loss of function of a gene is independent of the genetic background (*Dowell et al., 2010*).

We therefore collected a large and diverse panel of *E. coli* strains (N = 894), for which we measured growth across 214 conditions, as well as obtained the genomic sequences for the majority of the strains (N = 696). For each gene in each sequenced strain we calculated a 'gene disruption score' by evaluating the impact of non-synonymous variants and gene loss. We then applied a model that combines the gene disruption scores with the prior knowledge on conditional gene essentiality to predict conditional growth defects across strains. The model yielded significant predictive power for 38% of conditions having a minimum number of strains with growth defects (at least 5% of tested strains). We independently validated a number of causal variants with complementation assays, demonstrating the feasibility of precision genetic interventions. Since our predictions did not apply equally well for all conditions, we conclude that the set of conditionally essential genes has diverged substantially across strains. Overall, we anticipate that the *E. coli* reference panel presented here will become a community resource to address the multiple facets of the genotype to phenotype research, ultimately enabling the development of biotechnological and personalized medical applications.

## Results

### The phenotypic landscape of the *E. coli* collection

We have assembled a large genetic reference panel of 894 *E. coli* strains able to capture the genetic and phenotypic diversity of the species. These are broadly divided into natural isolates (N = 527) and strains derived from evolution experiments (N = 367). Out of these, 321 had available genome sequences, and we obtained the genomes of additional 375, reaching a total of 696 strains. The full list of strains, including name, collection of origin and links to relevant references is provided in *Supplementary file 1* and online at https://evocellnet.github.io/ecoref.

To test our capacity to develop genotype-to-phenotype predictive models we measured the fitness of this strain collection on a large variety of conditions (N = 214). We used high-density colony arrays and measured colony size as a proxy for fitness, similar to what has been applied before for the *E. coli* K-12 knockout (KO) library (*Nichols et al., 2011*). We used the deviation of each strain's colony size from itself across all conditions and all other strains in the same condition as our final

phenotypic measure (*Collins et al., 2006*). Thereby we obtained a list of conditions for which we know whether a tested strain has grown significantly less than the expectation (*Figure 1A* and Materials and methods). Both biological replicates (*Figure 1B*) and strains present in two distinct plates (*Figure 1—figure supplement 1*) were positively correlated (Pearson's r 0.693 and 0.648, respectively), indicating that we measured phenotypes with high confidence. Clustering of conditions based on phenotypic profiles across all strains was consistent with the conditions macro-categories (e.g. stresses *versus* nutrient sources) and with the number of sensitive strains (*Figure 1C*). The correlation of phenotypic profiles also clustered drugs with same mode of action (MoA), as shown by comparing the clusters' purity against those of random clusters (*Figure 1D*). The full phenotypic matrix for each strain across all conditions contains 114,004 single measurements (*Supplementary file 1*).

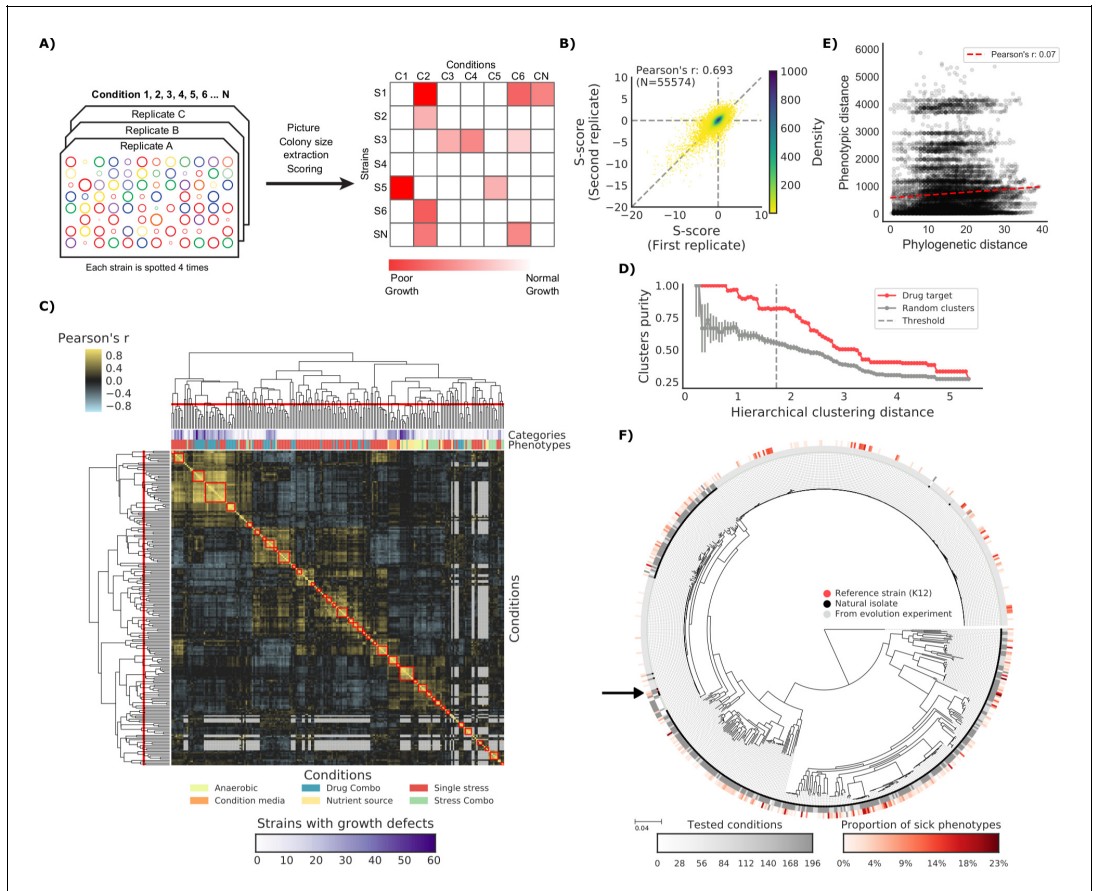

**Figure 1.** The phenotypic landscape of the *E. coli* strain collection. (**A**) Phenotypic screening experimental design and data analysis. (**B**) Phenotypic measurements replicability, as measured by pairwise comparing the S-scores of all three biological replicates. (**C**) Hierarchical clustering of condition correlation profiles; the threshold is defined as the furthest distance at which the minimum average Pearson's correlation inside each cluster is above 0.3. The two colored bands on top indicate the number of strains with growth defects for each condition and its category, showing consistent clustering. Gray-colored cells in the matrix represent missing values due to poor overlap of strains tested in the two conditions. (**D**) Clusters purity (computed for drug targets) for each hierarchical distance threshold, against that of random clusters (100 repetitions) shows that drugs with similar target tend to cluster together. (**E**) Pearson's correlation between phenotypic and phenotypic distances, based on phylogenetic independent contrasts (see Materials and methods). (**F**) Core genome SNP tree for all strains in the collection. Grey shades in the inner ring indicate the number of conditions tested for each strain, red shades in the outer ring indicate the proportion of tested conditions in which the strain shows a significant growth defect. The black arrow indicates the reference strain.

DOI: https://doi.org/10.7554/eLife.31035.002

The following figure supplement is available for figure 1:

**Figure supplement 1.** Phenotypic measurements reproducibility and correlation with genetic distance.

DOI: https://doi.org/10.7554/eLife.31035.003

If all genetic variants were to contribute to phenotypic divergence between strains we would expect to observe a strong positive correlation between those two metrics. This was not the case even when taking into account the phylogenetic dependencies between strains (r = 0.07, see Materials and methods). This finding reinforces the idea that most variants across these strains have a neutral impact on their phenotypes. Of particular interest are those strains derived from evolution experiments, such as the members of the LTEE collection (*Tenaillon et al., 2016*). While most strains (189 out of 266 tested) grew as expected in all tested conditions, a significant fraction (N = 77) exhibited at least one conditional growth defect phenotype. Again, the phylogenetic distance from the parental strains (REL606 and REL607) is not correlated with the proportion of growth phenotypes (Pearson's r: 0.08), even though hypermutators exhibit a slightly higher number of phenotypes (Cohen's d: 0.513). These results clearly underline that simple metrics of phylogenetic similarity are not predictive of differences in phenotype. Instead, few DNA variants are sufficient to cause clear phenotypic differences, indicating the importance of statistical or predictive strategies to prioritize those variants.

In toto, we have assembled an *E. coli* reference strain panel, which we have broadly phenotyped. This phenotyping resource recapitulates known biology, surveying a rich phenotypic space and providing insights into the evolution of phenotypes within the species.

## Gene level predictions of variant effects

As most DNA variants are likely to be neutral in their impact on gene function, variant effect predictors are required to build phenotype predictive models for each strain. For this we derived structural models and protein alignments covering 60.2% and 94.7% of the *E. coli* K-12 proteome, respectively (or 50.9% and 95.9% of all protein residues) and used them to compute the impact of all possible nonsynonymous variants (see Materials and methods, available at http://mutfunc.com). The impact of multiple non-synonymous variants within a gene was then combined using a probabilistic approach (*Jelier et al., 2011*), into a single likelihood measure of gene disruption (here termed 'gene disruption score', *Figure 2A* and Materials and methods). We also assumed that reference genes missing completely from a strain would have a high gene disruption score.

The gene disruption score can be considered as a relevant measure of the impact of mutations on gene function. Essential and phylogenetically conserved genes show lower average gene disruption scores across all strains, as compared to less conserved or random ones (*Figure 2B*). We also observed that genes that are predicted to lose their function together across all strains (*Figure 2C*) tend to be functionally associated (*Figure 2D and E*), in particular for genes belonging to the accessory genome (i.e. genes not present in all strains, *Figure 2—figure supplement 1*). These observations suggest that the gene disruption score is a biologically relevant measure of the impact of mutations at the gene level and that it could be used for growth phenotype predictions.

## Predictive models of conditional growth defects

We combined the gene disruption scores for each gene in each strain with the genes' conditional essentiality to obtain conditional growth predictive models. For each condition, we evaluated the impact of those variants affecting the genes that are essential for that same condition in the reference K-12 strain. We selected 148 conditions in which at least one strain displayed a growth defect and the *E. coli* K-12 KO collection had also been tested for (*Nichols et al., 2011* and Herrera-Dominguez et al., unpublished). For those, we computed a conditional score that would rank the strains according to their predicted growth level in the tested condition, from normal growth to the most defective growth phenotype (see Materials and methods). Briefly, if a strain has many detrimental variants in genes important for growth in a given condition, our predictive score will rank that strain as being highly sensitive in that condition. We then compared this predicted ranking with the experimental fitness measurements. The Area Under the Curve of a Precision-Recall curve (PR-AUC) was used for assessing our predictive power (*Figure 3A* and Materials and methods). We note that this predictive score is condition specific and no parameter fitting or training was used.

Our predictive score is able to discriminate strains with normal growth from ones with growth defects with significantly higher power than randomized scores. Both the predicted impact of single nucleotide variants and gene presence/absence patterns contribute to the predictive power of the model (*Figure 3—figure supplement 1*). The predictive power increases for conditions in which

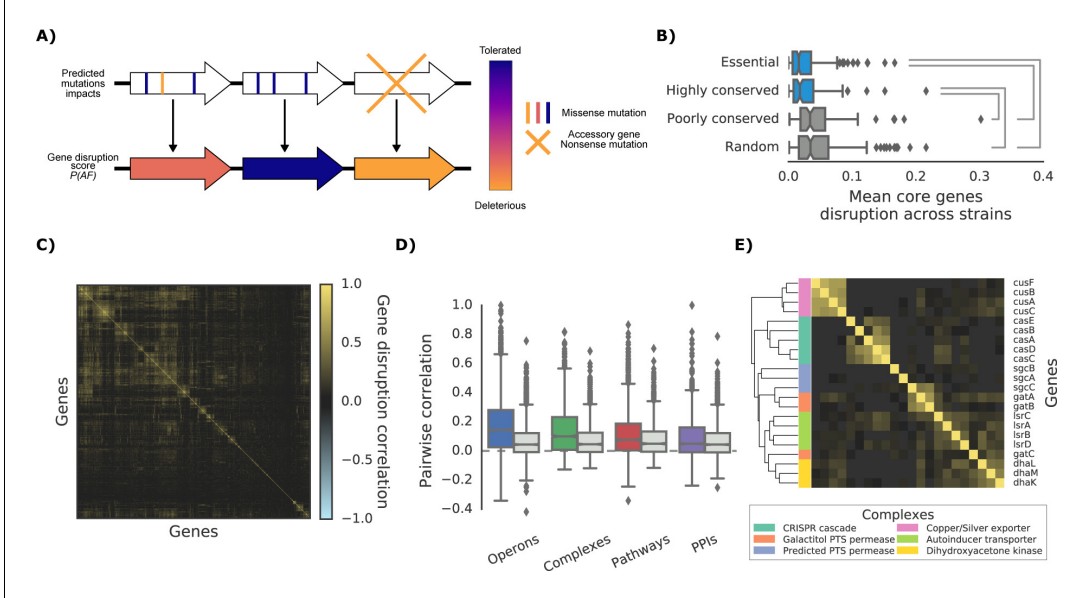

**Figure 2.** The 'gene disruption score', a gene-level prediction of the impact of genetic variants. (A) Schematic representation of how all substitutions affecting a particular gene are combined to compute the gene disruption score. (B) Average gene disruption score across all strains for four categories of genes conserved in all strains ('core genome'). Conserved genes are defined as those genes found in more (or less) than 95% of the bacterial species present in the eggNOG orthology database (*Huerta-Cepas et al., 2016a*). Statistically significant differences (Cohen's d value >0.3) are reported. (C) Gene-gene correlation profile of gene disruption across all strains shows clusters of potentially functionally related proteins. (D–E) The gene disruption profiles as a predictor of genes function. (D) Pairwise correlation of gene disruption scores inside each annotation set (colored boxes) and inside a random set of genes of the same size (grey boxes). (E) Gene-gene correlation profile of gene disruption scores in protein complexes; shown here a subset with high disruption score correlation.

DOI: https://doi.org/10.7554/eLife.31035.004

The following figure supplement is available for figure 2:

**Figure supplement 1.** Additional properties of the gene disruption score.

DOI: https://doi.org/10.7554/eLife.31035.005

more strains displayed growth defects (*Figure 3B*). We verified the significance of our predictions through the comparison against three randomization strategies: shuffled strains, shuffled gene sets and random gene sets (see Materials and methods). For all three strategies we observed a clear dependency between the PR-AUC value and the significance of our predictions (*Figure 3C* and *Figure 3—figure supplement 2*). Since some conditions are similar and share a significant fraction of their conditionally essential genes, we sometimes observe a skew in the performance of the 'shuffled set' randomizations (*Figure 3C*). We correctly predicted 20% of conditions that have at least 1% of poorly growing strains, with higher predictive capacity for conditions with larger number of strains with growth defects, reaching 38% for all conditions with more than 5% of poorly growing strains (*Figure 3—figure supplement 1*). Weighting the contribution of the conditionally essential genes to each condition also improved the predictive power, particularly for well-predicted conditions (*Figure 3—figure supplement 1*, Materials and methods).

To independently validate our predictive models, we carried out a GWAS analysis based on genes presence/absence and the growth phenotypes. Consistent with the validity of our models, we found that for conditions we predict with higher confidence (PR-AUC >= 0.1), there is a significant overlap between the K-12 genes predicted to be essential in the condition and the genes associated with poor growth by the association analysis (*Figure 3D*, Fisher's exact test, p-value 0.005).

We further examined two well predicted conditions (PR-AUC >0.35), pseudomonic acid 2 µg/ml (the antibiotic mupirocin) and minimal media with the addition of amino acid mix, to inspect our predictive model (*Figure 4*). Both conditions showed an enrichment of strains with growth defects at high predicted scores (GSEA p-values of 0.001 and $<10^{-6}$, respectively), which is a common property of conditions with higher PR-AUC (*Figure 3—figure supplement 1*). The reference K-12 strain

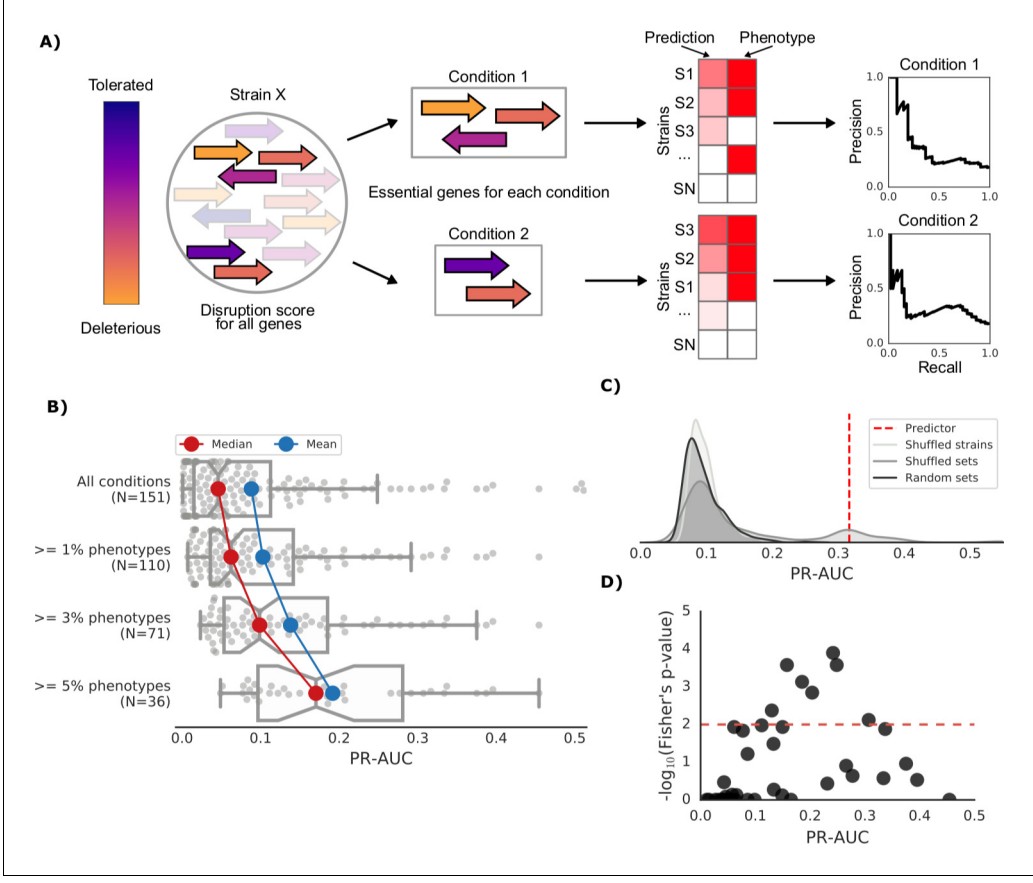

**Figure 3.** Prediction of growth-defect phenotypes in the *E.coli* strain collection. (**A**) Schematic representation of the computation of the prediction score and its evaluation; for each condition the predicted score is computed using the disruption score of the conditionally essential genes. The score is then evaluated against the actual phenotypes through a Precision-Recall curve. (**B**) Higher predictive power for conditions with higher proportion of growth phenotypes. For each condition set, the PR-AUC value for each condition is reported, together with the median and mean value (**C**) Significance of the PR-AUC value reported for the condition 'Clindamycin 3 μg/ml', against the distribution of three randomization strategies. 'Shuffled strains' indicates a prediction in which the actual strains' phenotypes have been shuffled; 'shuffled sets' indicates a prediction where the conditionally essential genes of a different condition have been used, and 'random set' indicates a prediction where a random gene set has been used as conditionally essential genes. For all three randomizations we report a significant difference between the actual prediction and the distribution of the randomizations (q-values of 1E-30, 0.05 and 1E-22, respectively). See *Figure 3—figure supplement 2* for the other conditions. (**D**) Genome-wide gene associations are in agreement with the predictive score; the enrichment of conditionally essential genes in the results of the gene association analysis is significantly higher in conditions with higher PR-AUC.

DOI: https://doi.org/10.7554/eLife.31035.006

The following figure supplements are available for figure 3:

**Figure supplement 1.** Detailed view of the predicted growth score and its properties.
DOI: https://doi.org/10.7554/eLife.31035.007

**Figure supplement 2.** Significance of the PR-AUC value reported for all conditions with at least 5% of the tested strains showing a sick phenotype, against the distribution of three randomization strategies.
DOI: https://doi.org/10.7554/eLife.31035.008

---

harbors many conditionally essential genes in minimal media (N = 181), providing an example for which growth phenotypes are well predicted from deleterious effects across a large number of genes in different strains. In contrast, pseudomonic acid is a condition with few (N = 10) conditionally essential genes, thus making it easier to evaluate the contribution of single genes to the phenotype. Of the 25 strains with highest predicted score in pseudomonic acid, 13 have been misclassified by the model. Seven misclassified strains had high disruption scores in either *lpcA* or *rfaE*, two genes

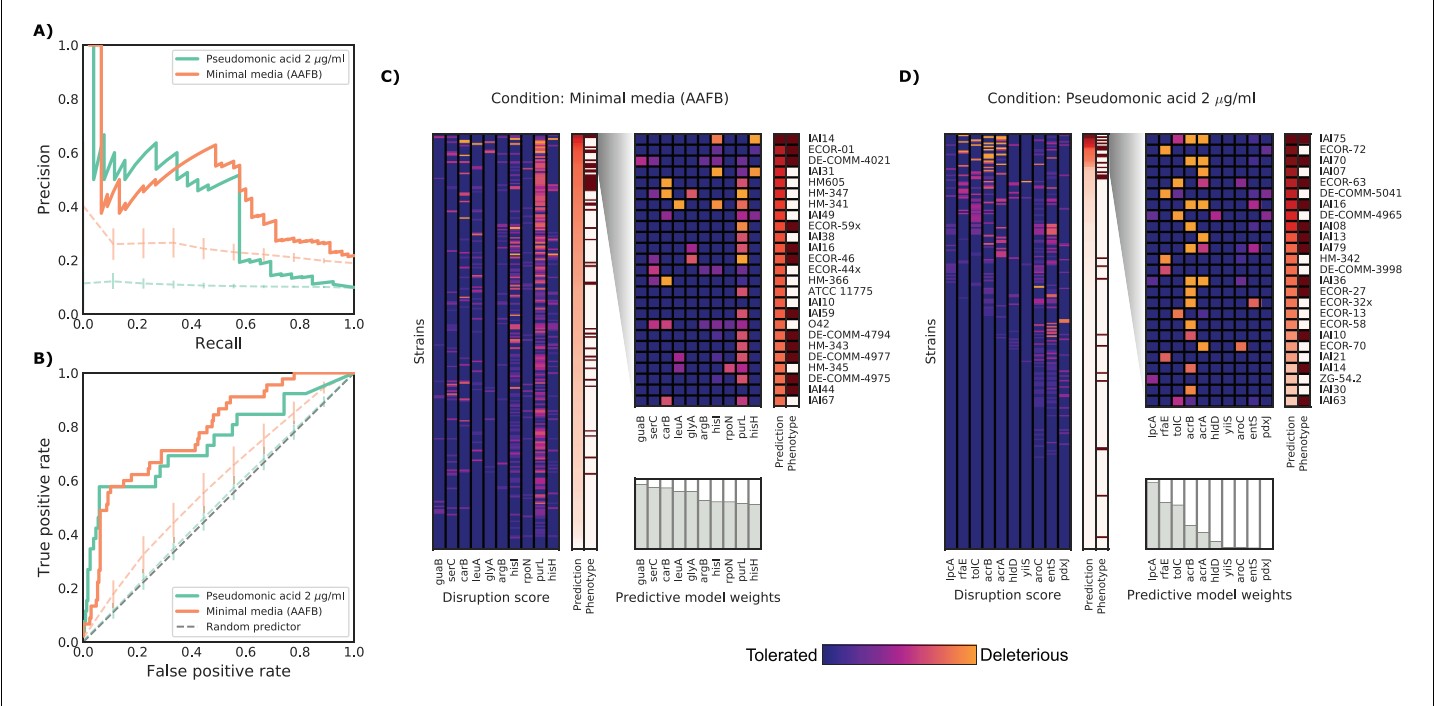

**Figure 4.** Detailed example on the computation and evaluation of the predicted score on two conditions. (**A**) Precision-Recall curve for the two example conditions. Dashed lines represent the average Precision-Recall curve for the same predictions carried out using 10'000 random gene sets of the same size of the actual conditionally essential genes for both conditions. Vertical lines represent the mean absolute deviation for the precision across the randomizations. The two randomization sets are significantly different than the actual predictions (q-values of 1E-33 and 0.02, respectively). (**B**) Receiver operating characteristic curve for the two example conditions. Dashed lines represent the same randomizations as in (**A**). (**C–D**) For each condition, the gene disruption score for the conditionally essential gene across all strains is reported, together with the resulting predicted conditional score and actual binary phenotypes (pale red: healthy and red: growth defect). Strains are sorted according to the predicted conditional score, while genes are sorted according to their weight in the predictive model; only the top 10 conditionally essential genes are shown. The inset reports the disruption score, predicted score and actual phenotypes for the top 25 strains.

DOI: https://doi.org/10.7554/eLife.31035.009

involved in the biosynthesis of a lipopolysaccharide (LPS) precursor (L-*glycero*-β-D-*manno*-heptose). Both, when deleted in K-12, cause a strong growth defect under pseudomonic acid. Only 1 of the correctly predicted strains had a mutation in one of those two genes (IAI36), suggesting that this pathway might be conditionally essential only in K-12. Another example of incorrect predictions involves two strains (ECOR-27 and ECOR-58) that share very similar disruption score profiles for the conditionally essential genes in pseudomonic acid (*Figure 4C*), but only ECOR-27 had a growth defect in this condition. Both strains harbor a single nonsynonymous mutation in *acrB* (E414G for ECOR-27 and I466T for ECOR-58), which, in both cases, is predicted as highly deleterious. Changes in conditional essentiality or epistatic effects are possible explanations for this misclassification, and therefore mapping and incorporating this information in our models could significantly benefit predictions in the future.

## Experimental validation of causal variants

Since we use mechanistic models in our phenotype predictions to calculate the impact of non-synonymous genetic variants, we can then also directly pinpoint the causal variants and implement genetic therapies to correct growth phenotypes. We tested this by ranking the mutated conditionally essential genes in each condition according to their predicted ability to rescue growth phenotypes (*Figure 5A* and Materials and methods). Several genes were predicted to restore many growth phenotypes (*Figure 5B*), including the genes forming the AcrAB-TolC multidrug efflux pump, which were predicted to restore growth in up to ~1000 condition-strain pairs (1012 *acrB*, 494 *acrA* and 517 *tolC*, respectively), reflecting the importance of this efflux system in drug resistance (*Li et al., 2015*).

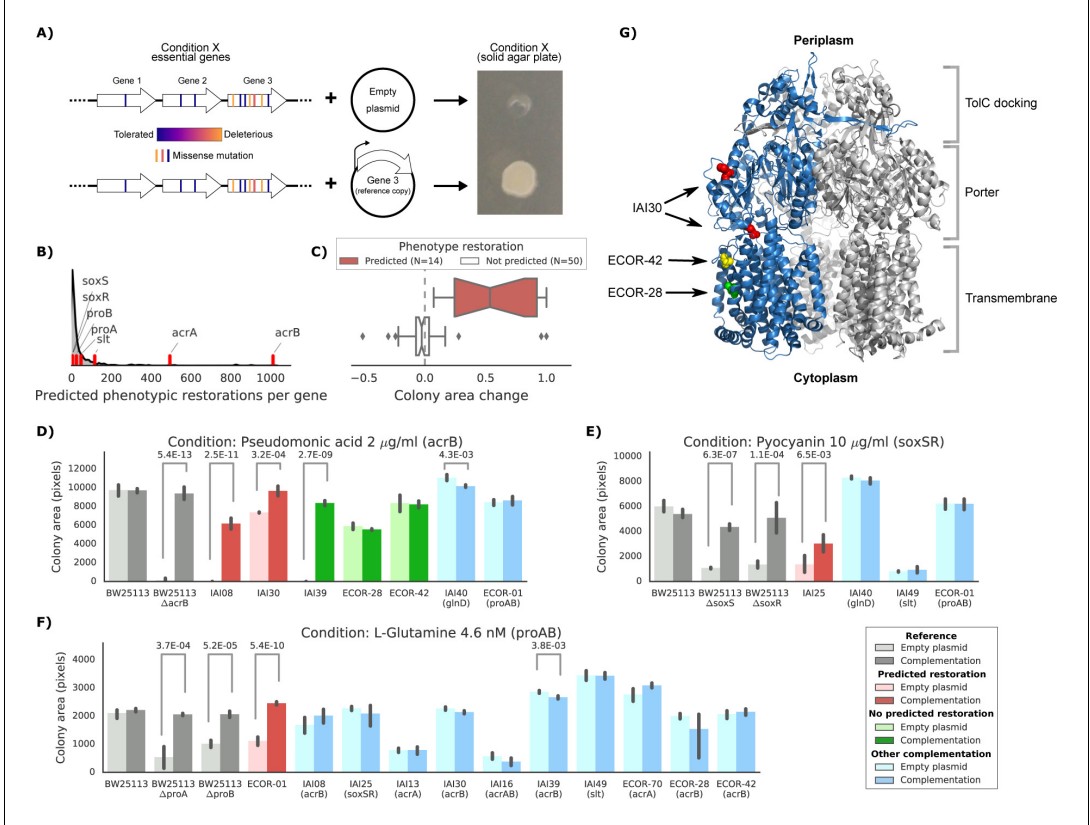

**Figure 5.** Experimental confirmation of the predicted phenotype-causing genotypes. (A) Schematic representation of the experimental approach. (B) Distribution of the number of predicted restored phenotypes per gene; red stripes indicate the genes that were experimentally tested. (C) Growth change between the target strain with the empty plasmid against the one expressing the reference copy of the target gene (complementation). Strains for which a change in phenotype is predicted are compared to those where no change is predicted. (D–F) Detailed representation of the results of the complementation experiment in three conditions. Mean colony area and the 95% confidence intervals are reported. Significant differences (t-test p-value<0.01) between colony area of the strains with the empty and complemented plasmids are reported. 'Other complementation' reports strains expressing a different gene than the focal one, indicated between parenthesis. (G) Cartoon representation of AcrB 3D structure (PDB entry: 2dr6). Only one of the three monomers is highlighted in blue; colored spheres represent known non-synonymous variants in the reported strains.
DOI: https://doi.org/10.7554/eLife.31035.010

The following figure supplement is available for figure 5:

**Figure supplement 1.** Detailed view of the experimental confirmation of the predicted phenotype-causing genotypes.
DOI: https://doi.org/10.7554/eLife.31035.011

We selected eight genes to experimentally verify our predictions, including genes involved in either drug resistance (*acrAB*, *soxSR* and *slt*) or auxotrophic growth (*glnD* and *proAB*). We then selected conditions for which the introduction of the reference allele in the target strains was predicted to restore growth, and controls where no improvement in phenotype was expected (Materials and methods). We also introduced the plasmid expressing the reference allele in the reference strain as a negative control and in the corresponding deletion strain from the KEIO collection (*Baba et al., 2006*) as a positive control. Overall, we observed a high correlation between replicates of this experiment (Pearson's r: 0.94, *Figure 5—figure supplement 1*). In total, we tested 64 strain-condition combinations, detecting a significant difference (p-value 8.4E-11, two-sided t-test) in colony size change between strain-condition pairs in which we had predicted growth will be restored (N = 14) versus ones we had not (N = 50) (*Figure 5C* and Materials and methods). This validated the effectiveness of our predictive models and more generally confirmed that genetic variants can be used to predict causal mutations and prioritise strategies for reverting/modifying phenotypes.

We investigated three conditions in more detail: pseudomonic acid 2 μg/ml with an *acrB* complementation, pyocyanin 10 μg/ml (a toxic secondary metabolite produced by *Pseudomonas*

*aeruginosa*) with a *soxSR* complementation and the L-glutamine amino acid as nitrogen source with a *proAB* complementation (*Figure 5D–F* and *Figure 5—figure supplement 1* for all conditions and strains). In all cases, strains harboring the gene(s) predicted to restore the phenotypes grew significantly better than strains carrying the empty plasmid (t-test p-value<0.01). In contrast, none of the strains harboring a different complementation gene showed a significant increase in colony size. In one case we detected an unexpected increase in colony size for one of the strains where we hadn't predicted any change (strain IAI39 expressing AcrB, *Figure 5D*). We hypothesize that the original growth phenotype is either due to an incorrectly classified variant in *acrB* or due to another variant that acts on the expression of this efflux pump.

Of the five strains expressing the reference allele of *acrB* tested in pseudomonic acid, three harbored nonsynonymous variants in their chromosomal copy. One of them (IAI30) was predicted to increase its colony size after the complementation, whereas the other two (ECOR-42 and ECOR-28) were not predicted to be affected by the complementation (*Figure 5D*). We mapped those nonsynonymous variants to the three-dimensional structure of AcrB (*Murakami et al., 2006*) to inspect their potential impact on the protein function (*Figure 5G*). Both ECOR-42 and ECOR-28 harbor a single deleterious nonsynonymous variant (SIFT, p-value~0.01) in the transmembrane domain of the protein (A915D and T1013I, respectively). Strain IAI30 on the other hand carries two nonsynonymous variants: E567V and H596N, both located in the AcrB PC1 subdomain, which is involved in the entry of the ligand that will be then extruded by the efflux pump (*Murakami et al., 2006*; *Seeger et al., 2006*). Since the E567V variant is predicted to be highly deleterious (SIFT, p-value~0.001), we presume that it impairs the fundamental drug uptake function of the AcrAB-TolC multidrug efflux pump, while the variants in the other two strains might not be significantly affecting the pump's function. This example shows how mechanistic interpretations of the impact of genetic variants can direct insights into the emergence of fitness defects and suggest potential gene therapeutic strategies, down to the level of the single genetic variant.

## Discussion

As part of this study, we have assembled and phenotyped a large, phylogenetically diverse *E. coli* resource strain collection. We observed little to no correlation between genotypic and phenotypic distance across strains. This demonstrates the need of assessing the impact of genetic variability as most variants are likely to be neutral. We combined these mechanistic interpretations of the impact of non-synonymous variants in the genomes of all these strains with prior knowledge on *E. coli* K-12 gene conditional essentiality to predict conditional growth defects of strains in the collection. This resulted in reliable predictions for up to 38% of the tested growth conditions. Our mechanistic-based model directly pinpoints the causal variants and allows us to design successful genetic therapies, something that is often the bottleneck in traditional GWAS approaches. Moreover, our approach opens up the door for predicting strain functional capacities in less studied microbes, such as those present in the human microbiome, with sole requirements being the strain genome sequence and knowledge of gene function in a model strain.

Despite the significant predictive power, we still could not reliably predict the strains phenotypic response in a number of conditions. Epistatic interactions, the impact of synonymous variants, variants in non-coding regions and the large accessory genome are four possible factors influencing phenotypic diversity. Models that account for these factors could therefore be required for more accurate predictive models. As compared to the previous application of this method to *S. cerevisiae* strains (*Jelier et al., 2011*), we observed a lower overall accuracy. The larger and more diverse strains cohort, the higher diversity in conditions tested, including several concentrations for each chemical, and the larger genetic diversity in bacteria are potential causes for the differences we have observed. Most importantly, we postulate that conditional essentiality may not be conserved across strains and therefore that it interacts with each specific genetic background. Previous work in two closely related *S. cerevisiae* strains and five human cancer cell lines has shown that gene-specific phenotypes can diverge considerably (*Ryan et al., 2012*; *Hart et al., 2015*). Given their higher genetic variability, this genetic background dependence of gene-specific phenotypes is expected to be stronger in *E. coli* and other bacteria, even though this remains to be tested. Despite these limitations we believe that we have demonstrated how gene function assays and genetic variant prioritization can be leveraged to deliver growth predictions and genetic intervention strategies. Future

iterations based on a more complete understanding on conservation of conditional essentiality should improve these predictions.

Finally, we propose this strain collection as a valuable community resource to develop and test genotype-to-phenotype models, following the example of previous successful genetic reference panels (*Ayroles et al., 2009*; *Liti et al., 2009*; *Bennett et al., 2010*; *Atwell et al., 2010*; *1001 Genomes Consortium, 2016*; *Weinstein et al., 2013*; *1000 Genomes Project Consortium et al., 2015*). Any additional molecular and phenotypic measurement on the collection members will amplify the benefit for the entire research community, moving us closer to understanding the basic biological question of how genetic variation translates to differences among individuals.

## Materials and methods

### Genome sequencing, assembly and annotation

Strains whose genome sequence were not yet available were sequenced using various Illumina paired-end platforms (*Supplementary file 1*). The resulting sequencing reads were quality checked using FastQC version 0.11.3, and contaminating sequencing adapters were removed using seq_crumbs version 0.1.9. Reads were assembled with Spades (*Bankevich et al., 2012*) version 3.5.0, using different k-mer sizes according to reads length and with the 'careful' option to reduce assembly errors; contigs below 200 base pairs were excluded. Resulting assembled contigs were annotated for coding genes, ribosomal RNAs and tRNAs using Prokka (*Seemann, 2014*) version 1.11. Strains not belonging to the *E. coli* species were excluded from subsequent analysis after being highlighted by Kraken (*Wood and Salzberg, 2014*) version 0.10.5. When available, typing information was used to spot incorrect genome sequences due to culture contaminations or other factors; known strains typing was compared to the ones predicted from the genome sequence using mlst version 2.8. Strain names were amended when possible (see the 'Notes' column in *Supplementary file 1*). The ECOR collection was carefully checked for inconsistencies, as it is well known that different 'versions' of this collection are circulating in the scientific community (*Johnson et al., 2001*; *Clermont et al., 2015*). The genome sequences were further checked for duplicated genomes: strains with highly similar genomes but highly divergent phenotypes (phenotypes S-score correlation below 0.6) were flagged and the genome belonging to the most likely incorrect strain was removed. If the conflict could not be resolved (i.e. using known typing information) both genomes were removed. Highly similar genomes were defined as those genomes whose distance was found to be below 0.001, as measured by mash (*Ondov et al., 2016*), version 1.1. Despite the draft status of the genomes presented in this study, we observed a similar core genome size as the one computed using a set of 376 complete *E. coli* genomes downloaded from NCBI's RefSeq (data not shown).

### SNP calling and annotation

Due to the variability in sequencing technologies or lack of the original reads for the already sequenced strains in the collection (N = 373), SNPs were called through a whole genome alignment between each strain and the genome of the reference individual (*Escherichia coli* str. K-12 substr. MG1655, RefSeq accession: NC_000913.3, strain collection identifier NT12001), using ParSNP (*Treangen et al., 2014*) version 1.2. Repeated regions in the reference genome were highlighted and masked through nucmer (*Kurtz et al., 2004*) version 3.1 and Bedtools (*Quinlan and Hall, 2010*) version 2.26.0. SNPs were then phased, and annotated using SnpEff (*Cingolani et al., 2012*) version 4.1g.

### Pangenome analysis

Genes present in the reference individual but absent in each strain were highlighted by computing hierarchical orthologous group using OMA (*Altenhoff et al., 2013*) version 1.0.6. Each strain was previously re-annotated using Prokka (*Seemann, 2014*) version 1.11 to harmonize gene calls.

### Phylogenetics

Strains phylogenetic tree was computed using a single ParSNP (*Treangen et al., 2014*) analysis, which generates a whole genome nucleotide alignment across all strains, which is then used as an input for FastTree (*Price et al., 2010*) version 2.1.7. The tree was visualized using the ete3 library (*Huerta-Cepas et al., 2016a*) version 3.0.0.

Phylogenetic distances between pairs of strains has been computed using phylogenetic independent contrasts, in order to account for phylogenetic interdependencies between strains (*Felsenstein, 1985*). In short, we have first restricted the phenotypic data to those strains and conditions with less than 10% missing data; we then imputed the rest using the average S-score for each strain, using the scikit-learn library (*Pedregosa et al., 2011*) version 0.17.1. For each condition we then computed the phylogenetic independent contrasts across the internal nodes of the strains' tree, using the ape, geiger and phytools libraries (*Paradis et al., 2004*; *Harmon et al., 2008*; *Revell, 2012*), versions 5.0, 2.0.6 and 0.6–20, respectively. The phenotypic distance between nodes was then computed using the euclidean distance between the phylogenetic independent contrasts across all conditions, using the nadist library version 0.1.0.

## Computation of all possible mutations and their effect

The impact of all possible nonsynonymous substitutions on the reference individual has been precomputed to speed up the lookup process. Functional impact of nonsynonymous substitutions has been computed using SIFT (*Ng and Henikoff, 2001*) version 5.1.1, using uniref50 as a sequence search database. Structural impact of nonsynonymous substitutions has been computed using FoldX (*Guerois et al., 2002*) version 4; both 3-D structures present in the PDB database and homology models were used. Homology models were created using ModPipe (*Pieper et al., 2014*) version 2.2.0. Conversion from PDB to Uniprot residues coordinates was derived from the SIFTS (*Velankar et al., 2013*) database. PDB structures were 'repaired' to fix residues' torsion angles and Van der Waals clashes before computing the impact of non-synonymous variants on structural stability. All the precomputed impacts of all possible nonsynonymous substitutions are available through the mutfunc database (http://mutfunc.com).

## Computation of the disruption score

For each strain and each protein coding genes we have computed the overall impact of all nonsynonymous and nonsense substitutions, in a similar approach as the one used for *Saccharomyces cerevisiae* (*Jelier et al., 2011*). The output of each predictor (a deleterious probability for SIFT and a $\Delta\Delta G$ value for FoldX) has been converted to the probability of the substitution being neutral $p(neutral)$. Mutations with known impact on the reference individual have been downloaded from Uniprot (*UniProt Consortium, 2015*, N = 3673) and used to derive such conversion; since only 580 mutations are reported to have a neutral impact, we added all observed nonsynonymous variants affecting known essential genes (as reported in the OGEE database, *Chen et al., 2017*) to the list of tolerated mutations. The distribution of the negative natural logarithm of the SIFT probability (plus a pseudocount equivalent to the lowest observed SIFT probability) for all the 6460 mutations was binned and the $p(neutral)$ was computed as the proportion of tolerated mutations over the total number of mutations in each bin. A logistic regression curve was then fitted to the binned distribution to derive the conversion between the SIFT probability and $p(neutral)$. For FoldX we used a similar approach, but using the computed $\Delta\Delta G$ value. The fitted logistic regression curves resulted in the following $P(neutral)$ functions:

$$P(neutral_{SIFT}) = \frac{1}{(1 + e^{-(0.625\log(SIFT + 1.527E^{-4}) + 1.971)})}$$

$$P(neutral_{FoldX}) = \frac{1}{(1 + e^{-(1.465FoldX + 1.201)})}$$

The $P(neutral)$ value attributed to nonsense mutations was assigned through heuristics: if the new stop codon was found within the last 16 residues of the protein it was given a $P(neutral)$ value of 0.99, reflecting its unlikelihood of disrupting the function of the protein, 0.01 otherwise. Losing a start or a stop codon was given a $P(neutral)$ value of 0.01, as they are very likely to impair protein function.

We gave a $P(neutral)$ value of 0.01 to those genes that were found to be present in the reference individual but absent in the target strain, reflecting the fact that their function is most likely to be absent from the target strain.

We inferred the probability that each gene had its function affected by the ensemble of the substitutions in each strain by computing a disruption score, equivalent to the $P(AF)$ (probability of the function being affected) used in *S. cerevisiae* study (*Jelier et al., 2011*).

$$P(AF) = 1 - \prod_{i=1}^{k} P_i(neutral)$$

Where *k* is the ensemble of nonsynonymous and nonsense substitutions observed in each gene. When both FoldX and SIFT predictions were available for a given substitution, we used the SIFT prediction only. Variants with relatively high frequency (>=10%) in the strains collection were not considered, as well as those reference genes that are absent in a significant number of strains (>=10%), as we reasoned that they are unlikely to be deleterious given their high observed frequency. Given that many strains of the collection are closely related (e.g. strains derived from the LTEE collection), we clustered them based on phylogenetic distance before applying the filtering. We also didn't consider variants and absent reference genes shared by all members of the LTEE strain collection, as those variants are present in the collection founder strain and therefore unlikely to affect the evolved clones phenotypes. Disruption scores for all proteins across all strains can be found in *Supplementary file 5*.

## Use of the disruption score as a functional association predictor

We used the proteins pairwise Pearson's correlation of disruption scores as a predictor of genes functional associations. We used four benchmarking sets: operons, as derived from the DOOR database (*Mao et al., 2014*), protein complexes and pathways derived from the EcoCyc database (*Keseler et al., 2013*), and protein-protein interactions derived from a recent yeast two-hybrid experiment (*Rajagopala et al., 2014*). The distribution of the disruption score correlation for each pair of related genes was compared against the same number of gene pairs randomly drawn from all reference genes. We also assessed the predictive power of the disruption score correlations by drawing a receiving operator characteristics curve (ROC) across each correlation threshold, using the scikit-learn library (*Pedregosa et al., 2011*) version 0.17.1.

## Strains phenotyping

The strains phenotypes were measured in a similar way as the *E. coli* reverse genetic screen (*Nichols et al., 2011*). The strain collection was plated in three solid agar plates, each one containing 1536 single colonies, so that each strain was plated at least four times in each plate, each time with different neighboring strains. For each condition we prepared three replicates (each one using a different source plate to reduce batch effects) with the concentrations indicated in *Supplementary file 2* and the addition of the Cosmos dye (CAS number 573-58-0) and Coomasie brilliant blue R-250 (CAS number 6104-59-2). The solution stains colonies when biofilm is being developed. Plates were stored in darkness at room temperature, unless otherwise required by the specific condition (i.e. higher temperature), and photographs of the plates were taken until colonies were found to be overgrowing into each other. Most of the conditions (N = 197) were tested at the same time and under the same laboratory conditions as the KEIO KO collection (Herrera-Dominguez et al., unpublished).

A series of colony parameters were extracted from each photograph, using Iris (*Kritikos et al., 2017*) version 0.9.4.71: colony size, opacity, roundness and color intensity. The most appropriate time point for each condition was determined by imposing a restriction on median colony size; between 1900 and 3600 pixels for the first two plates, and between 1300 and 3600 for the last plate, which contained the strains derived from evolution experiments, which tend to grow less than the natural isolates. The time points passing the first threshold were then sorted by the proportion of colonies with high roundness (>0.8), which is indicative of the overall quality of the plate, proportion of colonies over the minimum median colony size threshold, the spread of the colony size distribution (the lower the better), and mean colony size correlation across replicates.

A series of additional quality control measures were taken on the colony parameters. In order to remove systematic pinning defects, colonies appearing to be missing (colony size of zero pixels) in more than 66% of the tested plates were removed, unless all the internal replicates were found to be missing. Colonies with abnormal circularity were removed, as they were mostly due to incorrect

colony recognition by the software: colonies with size below 1000 pixels and circularity below 0.5 and colonies with size above 1000 pixels and circularity below 0.3 were removed. Putative contaminations were spotted and removed through a variance jackknife approach: first, the size of two outermost rows and columns colonies was corrected to match the median of the rest of the plate, then for each strain, each of the four replicates inside the plate was tested whether it contributed to more than 90% of the total colony size variance. If so, the replicate was flagged as a contamination and removed. The same approach was repeated using colony circularity, with a variance threshold of 95%.

The final colony sizes were used as an input for the EMAP algorithm (*Collins et al., 2006*), with default parameters, in order to derive an S-score, which informs on the deviation of each strain from the expected growth in each condition. Final S-scores were quantile-normalized, and significant phenotypes were highlighted using a 5% FDR correction similar to the one used in the *E. coli* reverse genetic screen (*Nichols et al., 2011*), using the statsmodels library version 0.6.1. The phenotypic data is available in *Supplementary file 4*.

## Computation of the conditional score and its assessment

Conditionally essential genes were derived for each condition of the *E. coli* reverse genetic screen overlapping with the conditions tested on the natural isolates collection. Mutants with a significant growth phenotype were considered to derive the list of conditionally essential genes. The conditional score for each strain, indicating the growth prediction, was computed as follows:

$$S_{s,c} = \sum_{g=1}^{l} \frac{1}{E_s} W_{g,c} \log(1 - P_{s,g(AF)})$$

Where *s* and *c* represent the strain and condition, respectively, *g* each conditionally essential gene for condition *c* (with size *l*), and $E_s$ representing a correction term for the disruption score, in order to remove the effect of phylogenetic distance (*Figure 2—figure supplement 1*).

$$E_s = \frac{1}{n} \sum_{g=1}^{n} \log(1 - P_{s,g}(AF))$$

Where *n* represents all the reference genes. The term $W_{g,c}$ is used to weight the contribution of each conditionally essential gene to the conditional score, and it is computed as follows:

$$W_{g,c} = -\log_{10}(F_{g,c}) \frac{C_g}{N_c}$$

Where $F_{g,c}$ is the FDR-corrected p-value of gene *g* in condition *c*, $C_g$ is the number of conditions in the chemical genomics screen where gene *g* shows a significant phenotype, and $N_c$ the total number of tested conditions in the chemical genomics screen.

The conditional score was assessed by computing a Precision-Recall curve, whose area (PR-AUC) was used as a direct measure of the predictive power of the method; the growth phenotypes were considered true positives. The PR curve and its AUC were computed using the scikit-learn library (*Pedregosa et al., 2011*) version 0.17.1. Three randomization approaches were used to generate control conditional scores: one using strains shuffling, one using conditionally essential gene sets shuffling, and one using random conditionally essential gene sets. Each randomization strategy has been used to generate 10,000 randomized scores, which were scaled to the actual one before assessment.

The conditionally essential gene sets, the conditional score and the PR-AUC values are available in *Supplementary file 5*.

## Association of accessory genes with phenotypes

Accessory genes from the strains collection pangenome were computed from the harmonized genome annotations made by Prokka (*Seemann, 2014*), using Roary (*Page et al., 2015*) version 3.6.1. The accessory genes were associated to each condition's phenotypes using Scoary (*Brynildsrud et al., 2016*) version 1.4.0, with default parameters. Genes with corrected p-value (Benjamini-Hochberg) of association below 0.05 were considered significant. The enrichment of

conditionally essential genes among the significant reference gene hits was assessed through a Fisher's exact test, as implemented in the SciPy library, version 0.17.0.

## Systematic in-silico complementation of conditionally essential genes

The potential to restore growth phenotypes through the introduction of reference alleles was predicted systematically in each strain by changing the disruption score to zero in each conditionally essential gene, and reporting the change in the conditional score : with respect to the maximum possible conditional score:

$$S_{s,c}^{max} = \sum_{g=1}^{l} \frac{1}{E_s} W_{g,c} \log(1 - P_{max}(AF))$$

Where $P_{max}(AF)$ is the maximum disruption score observed across all genes and strains. Any $\Delta S_{s,c}$ higher than 1% of $S_{s,c}^{max}$ was considered as potentially able to restore a growth phenotype.

## Experimental complementation of predicted phenotype-causing genes

In order to experimentally verify our predictions, we introduced the reference (BW25113) gene in a low copy plasmid. For the *slt* gene, we used the available plasmid from the TransBac library (H. Dose and H. Mori, unpublished resource, *Otsuka et al., 2015*). For *acrA*, *acrB* and *glnD*, we used the available plasmid from the mobile plasmid library (*Saka et al., 2005*). We amplified *acrAB*, *soxSR*, *proAB* from BW25113 and ligated into pNTR-SD (the backbone plasmid for the mobile plasmid library). Deletions of *acrAB*, *soxSR*, *proAB* in the reference strain were made using the lambda red recombination approach (*Datsenko and Wanner, 2000*). The resulting seven plasmids and the two empty plasmid controls were introduced into BW25113 (negative control), in the deletion strains from the Keio collection or constructed by us (positive control) and in the targets strains. All resulting strains were pinned using a Singer Rotor robot in 10 different conditions, on two solid agar plates, so that each strain is pinned at least four times per plate. The plates were incubated at room temperature and multiple photographs were taken until colonies were found to be overgrowing into each other. Iris (*Kritikos et al., 2017*) version 0.9.7 was used to extract colony size from the pictures.

## Code, data and strains collection availability

New genomic sequences have been deposited at the European Nucleotide Archive (ENA) with accession number PRJEB20550.

The source code used to perform the analysis reported here and generate the figures is available as *Source code 1* and at the following URLs: https://github.com/mgalardini/screenings, https://github.com/mgalardini/pangenome_variation, https://github.com/mgalardini/ecopredict (copies archived at https://github.com/elifesciences-publications/screenings, https://github.com/elifesciences-publications/pangenome_variation and https://github.com/elifesciences-publications/ecopredict). Code is mostly based on the Python programming language, and using the following libraries: Numpy (*Vanvan Derder Walt et al., 2011*) version 1.10.4, SciPy version 0.17.0, Pandas (*McKinney, 2010*) version 0.18.0, Biopython (*Cock et al., 2009*) version 1.68, scikit-learn (*Pedregosa et al., 2011*) version 0.17.1, fastcluster (*Müllner, 2013*) version 1.1.20, statsmodels version 0.6.1, PyVCF version 0.6.8, ete3 (*Huerta-Cepas et al., 2016a*) version 3.0.0, Matplotlib (*Hunter, 2007*) version 1.5.1, Seaborn (*Waskom et al., 2016*) version 0.7.1 and svgutils version 0.2.0.

Genomic and phenotypic data, as well as relevant information on how to obtain the members of the strain collection is available at the following URL: https://evocellnet.github.io/ecoref.

## Acknowledgements

We are particularly grateful to the various people providing us with many of the strains of this *E. coli* genetic reference panel, specifically (in alphabetical order): Alfredo G Torres, Catharina Svanborg, David Clarke, Erick Denamur, Ewa Bok and Pawel Pusz, Isabel Gordo and Lilia Perfeito, Jorg Weinreich and Peter Schierack, KC Huang, Lisa Nolan, Mark Goulian, Mathew Upton, Olin Silander, Richard Lenski, Scott Hultgren and Wanderley Dias da Silveira. We thank Amanda Miguel for helping in the phenotypic screen. We also thank the EMBL Gene Core, and especially Rajna Hercog and

Vladimir Benes for the support in genome sequencing. We thank Ewan Birney, Oliver Stegle, KC Huang and Zam Iqbal for critical reading of the manuscript. This work was partially supported by the Sofja Kovaleskaja Award of the Alexander von Humboldt Foundation to ATy and a grant from the Fondation pour la Recherche Médicale (Equipe FRM 2016, DEQ20161136698) to ED.

## Additional information

### Funding

| Funder | Grant reference number | Author |
|---|---|---|
| Alexander von Humboldt-Stiftung | Sofja Kovaleskaja Award | Athanasios Typas |
| Fondation pour la Recherche Médicale | Equipe FRM 2016, DEQ20161136698 | Erick Denamur |

The funders had no role in study design, data collection and interpretation, or the decision to submit the work for publication.

### Author contributions

Marco Galardini, Conceptualization, Formal analysis, Investigation, Methodology, Writing—original draft, Project administration, Writing—review and editing; Alexandra Koumoutsi, Lucia Herrera-Dominguez, Juan Antonio Cordero Varela, Anja Telzerow, Omar Wagih, Morgane Wartel, Investigation; Olivier Clermont, Resources, Data curation, Validation; Erick Denamur, Resources, Validation, Writing—review and editing; Athanasios Typas, Conceptualization, Supervision, Funding acquisition, Writing—original draft, Writing—review and editing; Pedro Beltrao, Conceptualization, Supervision, Writing—original draft, Project administration, Writing—review and editing

### Author ORCIDs

Marco Galardini http://orcid.org/0000-0003-2018-8242
Alexandra Koumoutsi http://orcid.org/0000-0001-8368-4193
Juan Antonio Cordero Varela http://orcid.org/0000-0002-7373-5433
Pedro Beltrao http://orcid.org/0000-0002-2724-7703

### Decision letter and Author response

Decision letter https://doi.org/10.7554/eLife.31035.023
Author response https://doi.org/10.7554/eLife.31035.024

## Additional files

### Supplementary files

• Source code 1. code used to analyze the genetic and phenotypic data, predict phenotypes and draw all the manuscript figures.
DOI: https://doi.org/10.7554/eLife.31035.012

• Supplementary file 1. detailed list of the members of the strain collection
DOI: https://doi.org/10.7554/eLife.31035.013

• Supplementary file 2. detailed list of conditions on which the strain collection has been tested
DOI: https://doi.org/10.7554/eLife.31035.014

• Supplementary file 3. categorization and mode of action (MoA) of each chemical used in the phenotypic screening
DOI: https://doi.org/10.7554/eLife.31035.015

• Supplementary file 4. s-scores, FDR corrected p-values and binary growth defect matrix of the E. coli phenotypic screening (869 strains across 214 conditions)
DOI: https://doi.org/10.7554/eLife.31035.016

• Supplementary file 5: .disruption scores, conditional scores and prediction assessments.
DOI: https://doi.org/10.7554/eLife.31035.017

• Transparent reporting form
DOI: https://doi.org/10.7554/eLife.31035.018

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
