## [Decision Letter]

Thank you for submitting your article "Phenotype inference in an *Escherichia coli* strain panel" for consideration by *eLife*. Your article has been reviewed by two peer reviewers, and the evaluation has been overseen by Naama Barkai as the Senior Editor and Reviewing Editor. The reviewers have opted to remain anonymous.

The reviewers have discussed the reviews with one another and the Reviewing Editor has drafted this decision to help you prepare a revised submission.

As you will see below, the reviewers appreciated the extensive analysis and believe that the data will be a useful resource for the community. Please see below technical issues that were raised by the reviewers, which should be addressed in full, as you revise your submission for the Tools and Resources section.

*Reviewer #1:*

This paper does an extensive analysis and prediction of phenotypes caused by mutations in a large *E. coli* strain panel evaluated over several hundred conditions.

Overall, my concern with this paper applies similarly to many large-scale, high-throughput works I see: The authors have done so much, and are trying to present so many different results and approaches in one paper, that it's difficult to evaluate what exactly was achieved. This paper is either a monumental achievement or a collection of trivial results, and I find it difficult to distinguish between the two possibilities.

I am tentatively willing to give the authors the benefit of the doubt, but I have specific questions and concerns that would strengthen my confidence in the work.

- "We detected no strong positive correlation" this is a strange sentence, in particular if it then lists a significant negative correlation. I think the authors need to explain first why they would expect a positive correlation.

Also, with regards to these correlations, because of the phylogenetic dependency among strains they should be calculated on phylogenetic independent contrasts. Alternatively, it would be better to not report a p value than to report an incorrect one.

- Figure 3: The precision-recall curves don't look particularly impressive. They rapidly decay to a precision below 0.5, at which point more cases are called incorrectly than correctly.

- Figure 3: This figure is not properly explained. First, what are the small gray dots? Second, what are the three thick gray dots labeled as shuffled strains, shuffled sets, and random sets? Each randomization should provide a distribution which then is compared to one PR-AUC value. If the true PR-AUC value is rare under reshuffling, then it is significant. This relationship is not in any way visible in the figure.

- Figure 4: Again, the precision-recall curve looks pretty bad. I grant that there's some enrichment of phenotypes towards the top end of the predicted phenotype scale, but the prediction quality is not impressive and it's not clear to me whether a trivial predictor (e.g., one only evaluating whether an amino acid has been changed towards a highly biochemically dissimilar one) could do just as well.

- Section on "Experimental validation of causal variants": It is not entirely clear to me what has been achieved here. If a mutation knocks out an essential gene then clearly that defect can be rescued via complementation. So the achievement would have to be to predict that a gene is essential only under specific conditions. Is that what was done? And if so, how? Was essentiality simply observed, by noting that strains with mutations in certain genes wouldn't grow under specific conditions? Or was it somehow inferred from some other data?

- "with prior knowledge on gene essentiality" I'm not entirely convinced that gene essentiality matters all that much. A mutation in a non-essential gene can have a substantial fitness effect, and a mutation in an essential gene that doesn't fully disable that gene's function may have only a minor fitness effect.

- It is unclear to me whether the code underlying the mutfunc database has been provided or not. The database looks nice, but I'm generally wary of any bioinformatics tool that I can't download and run on my own machine. The authors provide several pipelines (https://github.com/mgalardini/screenings, https://github.com/mgalardini/pangenome_variation, https://github.com/mgalardini/ecopredict), but none seems to perform the task of the mutfunc database.

*Reviewer #2:*

We have read the study entitled "Phenotype inference in an *Escherichia coli* strain panel" by Galardini et al. This study performs phenotypic screens for 894 *E. coli* strains across 214 growth conditions. Most of the strains (696) have full genome sequence available. Gene-level functional predictions were correct in up to 38% of cases. This dataset will become an important and useful resource for the systems biology and microbiology community. We would appreciate of the authors addressed our minor comments below:

We commend the authors for putting all of their data in easily accessible format on github with links out to all of the available assemblies and phenotype measurements. This inclusion will ensure that the resource is easy to use for the research community. However, one question remains regarding the new annotations for each strain. Are these available anywhere? It was difficult to find them both on the web or in the authors’ supplemental files.

The authors previously used this approach on yeast with a median predictive performance of 0.76 (Jelier et al. 2011), but were only able to reliably predict 38% of the conditions tested in this instance. Could the authors comment on why their performance was lower?

Why did the authors use ParSNP to compute phylogeny rather than other methods (such as MLST, whole genome alignment, etc). The authors look for a correlation between phylogeny and genotype. Could the lack of correlation be due to assumptions made when calculating the phylogeny?

Why did the authors choose to incubate the most of the conditions at room temperature rather than the more common 37C? Might this significantly impact their results (i.e. would growth capabilities change if incubated at 37C?).

How do the authors define the "accessory genes/genome". Are these simply all genes that are not part of the core genome? Could the authors comment on the completeness of the genomic sequences in the data set?

Were SNPs only called for sequences in common with the reference E. coli K12 genome? Would that potentially explain some of the discrepancies between the model predictions and the experimental outcome?

Does the term "mechanistic" appropriately describe the method used? Especially when one considers disruption scores in many areas of a genome and the conditional essentiality of genes?

---

## [Author Response]

Reviewer #1:

This paper does an extensive analysis and prediction of phenotypes caused by mutations in a large E. coli strain panel evaluated over several hundred conditions.Overall, my concern with this paper applies similarly to many large-scale, high-throughput works I see: The authors have done so much, and are trying to present so many different results and approaches in one paper, that it's difficult to evaluate what exactly was achieved. This paper is either a monumental achievement or a collection of trivial results, and I find it difficult to distinguish between the two possibilities.

We can understand the reviewer’s reluctance towards large-scale projects and of course we hope to convince him/her that this is not simply a large collection of data. We are in fact trying to ask a very fundamental biological question: given all that we currently know about *E. coli* biology, can we predict how a specific strain of *E. coli* will grow in a given condition based on its genome? It is not trivial at all to be able to take a genome sequence of an *E. coli* strain and determine in which conditions this strain is going to grow poorly in. By asking this question we are testing the limits of our knowledge of how DNA variation affects phenotypes. Working towards these predictions implies the capacity to encapsulate in computational models all of this understanding. We upfront admit that we are not yet fully capable of making such predictions, but we also think it is very important to test these limits and to challenge ourselves and the scientific community to improve on what we have achieved in this manuscript.

In particular, we have used the knowledge on *E. coli* gene function as defined by the chemical genomic screens done in one particular strain, the lab *E. coli* K-12. We then make the assumption that a gene that is essential for growth in a given condition in the K-12 strain will also be essential for growth in the same condition in any other strain of *E. coli*. Based on this knowledge we then compute the effects of non-synonymous variants from genomes of different strains, using folding and sequence conservation algorithms. This allows us to predict which strains will have growth defects in specific conditions. Our predictions are significantly better than random and we can show in the complementation assays that we are in fact predicting the genes that are causing the growth defects in the right conditions. Of course, there is much room for improvement and many ways to try to extend the predictions made here as we discuss in the manuscript. For example, we have not yet taken into account mutations that could alter the expression of the genes. As we and others incorporate such additional information the predictive models will become even more accurate. At the same time, there is also great opportunity for new biology from the deficiencies in the predictions. As we mentioned in the manuscript we see cases where loss of function of a gene is known to cause a condition specific growth defect in *E. coli* K-12 but the same is not observed in specific strains. We and others can now try to understand why this is the case.

Finally, in order to ask this fundamental question we also needed the genomes of many strains of *E. coli* and their growth phenotypes across a large number of conditions as well. This large-scale data gathering effort is in itself useful as a resource, but we would like to re-emphasize that the main focus of this paper is on the fundamental biological question we are asking.

I am tentatively willing to give the authors the benefit of the doubt, but I have specific questions and concerns that would strengthen my confidence in the work.- "We detected no strong positive correlation" this is a strange sentence, in particular if it then lists a significant negative correlation. I think the authors need to explain first why they would expect a positive correlation.Also, with regards to these correlations, because of the phylogenetic dependency among strains they should be calculated on phylogenetic independent contrasts. Alternatively, it would be better to not report a p value than to report an incorrect one.

We agree with the reviewer that we have failed in properly explaining what our expectations were with respect to this analysis. The message we wanted to convey was that if all genetic variation contributed equally to phenotypic variation we would observe a strong positive correlation between these two variables, which is not the case. The overall negative correlation presented in the main text is present merely because of the highly genetically similar strains coming from the LTEE collection (Long Term Evolution Experiment). Given this confounding factor and the suggestion to use phylogenetic independent contrasts to account for the strains’ dependencies, we have changed Figure 1 (now Figure 2). As suggested by the reviewer we are now deriving the correlation between genetic and phenotypic distance using the phylogenetic independent contrasts computed along the core genome tree of the strains. We show that we did not observe a positive correlation between genetic and phenotypic distance (Pearson’s r: 0.07), confirming the assumption that not all genetic variation contributes equally to differentiate phenotypes. We have also updated the Materials and methods section (“Phylogenetics”) to indicate how this analysis has been performed. We have also removed the last two panels of Figure 1—figure supplement 1.

- Figure 3: The precision-recall curves don't look particularly impressive. They rapidly decay to a precision below 0.5, at which point more cases are called incorrectly than correctly.

We thank the reviewer for raising this point. While the curves displayed in Figure 3 are merely part of the schematic used to explain the prediction and scoring strategies, they mimic some of the real curves we observed in our data, and therefore this comment is relevant and important. We address the concerns of the reviewer with respect to the drop in precision and general aspect of the curves in the comment about Figure 3 and Figure 4. In a nutshell, we would like to point out that the predictions are indeed significantly better than the various randomization strategies we have proposed.

- Figure 3: This figure is not properly explained. First, what are the small gray dots? Second, what are the three thick gray dots labeled as shuffled strains, shuffled sets, and random sets? Each randomization should provide a distribution which then is compared to one PR-AUC value. If the true PR-AUC value is rare under reshuffling, then it is significant. This relationship is not in any way visible in the figure.

We agree with the reviewer that the visualization we chose doesn’t provide sufficient information and, most importantly, hides the relationship between the distribution of the actual predictions versus the three randomization strategies we have adopted. We therefore decided to remove the thick gray dots that corresponded to the median results of the randomizations (“shuffled strains”, “shuffled sets”, and “random sets”) and leave the small grey dots, which represent the PR-AUC value of each condition separately. To better illustrate the randomisations we have added a panel to Figure 3 (panel C), showing an example condition (Clindamycin 3 μg/ml). As suggested by the reviewer we now show the distribution of the randomizations and in a red line the result for the actual predictor for this condition. In the figure legend we have indicated the q-value related to the likelihood that the actual predictions has been drawn from the distribution of each randomization. Furthermore, we have added a new supplementary figure (Figure 3—figure supplement 2), which shows for each condition a similar plot as the one presented in Figure 3. We have also updated the Results section to better explain this point. We believe that these changes will help the readers understanding why the predictions presented in this work are relevant.

- Figure 4: Again, the precision-recall curve looks pretty bad. I grant that there's some enrichment of phenotypes towards the top end of the predicted phenotype scale, but the prediction quality is not impressive and it's not clear to me whether a trivial predictor (e.g., one only evaluating whether an amino acid has been changed towards a highly biochemically dissimilar one) could do just as well.

The points raised by the reviewer are important. Indeed, the precision in the two example conditions drops at relatively low recall values. It is important to point out that the number of strains with growth defects in a given condition are a small fraction of the total and it is therefore a very unbalanced set of true and false positives. Achieving high precision is particularly challenging in such an unbalanced set. Nevertheless the rankings we predict are noticeably better than expected by random chance. Author response image 1 shows the comparison against the predicted score computed using random conditionally essential gene sets. In that respect the comments about Figure 3 allowed us to make this significance more evident.

Finally, we would like to point out how the choice of showing a precision-recall curve as the sole visual indicator of our prediction accuracy may be actually undermining the value of our approach. Author response image 1 shows two additional metrics for the two example conditions presented in Figure 4 (the “bootstrap” curves representing the average performance of 100 predictions using random gene sets):

As evident from the plots in Author response image 1, the predictions made in these two conditions are significantly better than using random gene sets. This is especially obvious in the ROC curve, where the randomizations are very similar to a random predictor (ROC AUC ~= 0.5), as opposed to very high ROC AUC for both conditions for the “real” predictions (~0.76). We have now updated Figure 4 to show how the predictions are significantly different than the “random sets” bootstraps. We have also added a new panel to Figure 4 to include the ROC curve.

While we recognize that our predictions are far from perfect, we believe they represent an important demonstration that is indeed possible to combine chemical genomics data from the reference K-12 strain with mutation data to infer conditional growth defects for other strains. In particular, leveraging the information on condition-dependent essentiality is key to the significance of our predictions; if we were to only use the predicted impact of non-synonymous variants we would not be able to deliver condition specific predictions. We have expanded our Discussion to point out this important conclusion of our study.

The reviewer asked if the same predictions could be achieved with a “trivial” variant predictor such as one that evaluates the effect of mutations based on amino-acids categories. Our predictions are a combination of variant effect predictions with gene function information from the K-12 reference strains. We assume that the reviewer is here asking about how much information is added by the interpretation of the variant effects only, keeping the gene function information unchanged. We have tested a more trivial variant effect predictor for the impact of nonsynonymous substitutions as suggested by the reviewer, and as can be seen from the three plots in Author response image 2, its predictive power is inferior to the one we propose.

**Author response image 2. respfig2:** 

In short, we have divided the 20 amino acids into 8 categories: hydrophobic (A, I, L, M, V), hydrophobic (aromatic) (F, W, Y), polar (N, Q, S, T), acidic (D, E), basic (R, H, K), cysteine (C), proline (P) and glycine (G). We then have assigned a P(neutral) value of 0.99 (tolerated) for amino acid substitutions within each category, and 0.01 (deleterious) otherwise. Again, all the other predictors (accessory genes and stop codons) have been kept the same as our proposed predictor, including the use of conditionally essential genes. The two top plots show the precision recall curves for 2 conditions comparing variant effect prediction approaches. The boxplots in the bottom figure summarize the results across all conditions that have at least 5% of strains with significant growth defects. We believe that this test demonstrates that there is added value in using variant effect predictors that take into account conservation and structural information as we are able to achieve higher precision. We have decided not to include this result in the main text, as introducing an additional predictor might generate confusion in the reader.

Finally, as indicated in the Discussion, we of course concede that these predictions have room for improvement, and suggest future directions on how to do so.

- Section on "Experimental validation of causal variants": It is not entirely clear to me what has been achieved here. If a mutation knocks out an essential gene then clearly that defect can be rescued via complementation. So the achievement would have to be to predict that a gene is essential only under specific conditions. Is that what was done? And if so, how? Was essentiality simply observed, by noting that strains with mutations in certain genes wouldn't grow under specific conditions? Or was it somehow inferred from some other data?

Looking at this and the subsequent comment by the reviewer, we suspect that we might have failed in properly explaining the approach we have taken and the terminology we are using. We therefore have updated the Introduction, Results and Discussion sections to make it easier to understand what exactly our approach consists of. In short, we have used the information on gene *conditional* essentiality derived from pre-existing chemical genomics screens from the *E. coli* lab strain K-12. We therefore know which genes are essential for growth in K-12 in a given condition, and we can therefore focus on interpreting genetic variation on those genes only, as they would be more likely to influence growth if a mutation has an impact on its function in another *E. coli* strain. This assumes that a given gene that is essential for growth in a condition in the *E. coli* strain K-12 will still be equally important for that same condition in any other strain of *E. coli*. As we described in the Discussion one of the potential reasons why we have achieved only moderate success could very well be because the genes essential for growth under specific conditions in the K-12 strain may not be essential for growth in those same conditions for other strains. While we have not proved this in this study, this would be an extremely important finding that would highlight the plasticity of gene function.

Going back to the experimental validation of our model, based on the mutations that we observed in the genome of a given strain we can then predict: 1) if a mutation will severely impact on the function of a giving gene; 2) if that gene is known to be essential for growth in a specific condition (in K-12). We can then predict a strain that has a damaging mutation in a gene essential to grow in a certain condition should also grow poorly in that condition. Purely from the genome sequence of a strain we can then rank all mutated genes as the most likely mutated gene that is causing a grown defect in a specific condition. These sets of complementation experiments show that we can in fact predict why a given strain of *E. coli* is growing poorly in a specific condition.

This is the reason why we believe we have demonstrated the long-term potential of our approach to pave the way for precision genetic intervention strategies. Our approach is specific and, to a certain extent, accurate. We believe that Figure 5 is an appropriate demonstration of this statement; in the experiments we have carried out we observe a clear difference between those complementations that were predicted to restore growth in specific conditions versus those that were not predicted to have an impact.

- "with prior knowledge on gene essentiality" I'm not entirely convinced that gene essentiality matters all that much. A mutation in a non-essential gene can have a substantial fitness effect, and a mutation in an essential gene that doesn't fully disable that gene's function may have only a minor fitness effect.

We believe that we have addressed this concern in the reply to the previous question. We indeed believe that *conditional* essentiality (i.e. essential for a given condition) matters, and we believe to have demonstrated it when showing how our predictions compare to the randomisations with shuffled and random gene sets (Figure 3, Figure 4 and Figure 3—figure supplement 2). Again, we thank the reviewer for pointing out how we have failed in making these points clear. We hope that our revised text improves how we explain this key concept.

- It is unclear to me whether the code underlying the mutfunc database has been provided or not. The database looks nice, but I'm generally wary of any bioinformatics tool that I can't download and run on my own machine. The authors provide several pipelines (https://github.com/mgalardini/screenings, https://github.com/mgalardini/pangenome_variation, https://github.com/mgalardini/ecopredict), but none seems to perform the task of the mutfunc database.

The mutfunc database contains all the precomputed non-synonymous substitutions and their predicted impact using a variety of computational approaches. As we described in the manuscript it includes predictions made with SIFT and FoldX which are tools that are available as standalones. A lot of work on our part went into running the calculations but the database itself is a front-end for all of the precomputed scores that we have obtained after many days of total compute time. No actual computation is done by the mutfunc server as all possible variants along the genome are already pre-computed. For these reasons there is no standalone tool as such that we can create. As the reviewer may imagine, the results of the pre-computed consequences of all possible variants along the genomes of the 3 species currently in mutfunc are also not something that can be easily provided as a standalone tool. The webservice we provide in mutfunc is not limited to searching small numbers of variants. Full genome variants can be queried with a very fast response time, since it is essentially looking up the pre-computed scores. There is also no limit in getting the outputs of large numbers of queries. We have added a few more details in the Materials and methods section (“Computation of all possible mutations and their effect”) regarding how we have run Sift and FoldX. If the reviewer found specific limitations from the webserver we can try to address them. We did try very hard to make all of the work associated with this publication available even before the work is published. To further facilitate the re-use of our results, we have now added the SIFT and FoldX scores for all the nonsynonymous variants observed in the strain collection to the Ecoref website, in the download section (https://evocellnet.github.io/ecoref/download/).

Reviewer 2:

[…] We commend the authors for putting all of their data in easily accessible format on github with links out to all of the available assemblies and phenotype measurements. This inclusion will ensure that the resource is easy to use for the research community. However, one question remains regarding the new annotations for each strain. Are these available anywhere? It was difficult to find them both on the web or in the authors’ supplemental files.

We also believe that making this resource easy to use for the community is a crucial point to our manuscript and long-term vision. We recognize that the gene annotations are important for the use of this resource and we therefore thank the reviewer to raise this point. We have released the annotated genomes in the ecoref website (“strains” section). The annotation for each strains are available in the standard and popular GFF3 format.

The authors previously used this approach on yeast with a median predictive performance of 0.76 (Jelier et al. 2011), but were only able to reliably predict 38% of the conditions tested in this instance. Could the authors comment on why their performance was lower?

If we understood the reviewer’s comment correctly, there might have been a misunderstanding. We are not the authors of the 2011 yeast paper (Jelier et al., 2011). As we highlighted in the Introduction, Jelier and coauthors have applied their predictive approach to a much smaller cohort of 15 strains across 20 conditions, most of which are related to metabolism. We therefore postulate that one potential reason for our lower performance when compared to the previous study is the larger variety in the type of conditions in our screen, but also in their strength. In fact some chemicals have been tested with multiple concentrations (e.g. amoxicillin), some of which might be suboptimal to elicit a strong response from the tested strains, making a prediction potentially more difficult. Another reason may be the larger genetic diversity in bacteria in general, and of our library in particular. We have added a short sentence in the discussion to explain the potential causes of these differences.

Why did the authors use ParSNP to compute phylogeny rather than other methods (such as MLST, whole genome alignment, etc). The authors look for a correlation between phylogeny and genotype. Could the lack of correlation be due to assumptions made when calculating the phylogeny?

We believe that we might have failed to properly explain the procedure we followed to obtain the strains’ phylogeny. As the reviewers suggests, a whole genome alignment is one of the best way to obtain a phylogeny. In fact, this is exactly what ParSNP does: it aligns the whole genome of the input genomes and it then feeds such alignment to FastTree (Price et al., 2010) which computes the final phylogeny. We have updated the methods section to explicitly state how the method works.

Why did the authors choose to incubate the most of the conditions at room temperature rather than the more common 37C? Might this significantly impact their results (i.e. would growth capabilities change if incubated at 37C?).

We did the strain collection phenotyping together with that of the KEIO gene-knockout collection – latter is part of a manuscript currently in preparation (Herrera-Dominguez et al.). We opted for room temperature because then we could simultaneously measure two readouts: growth and biofilm formation. *E. coli* forms biofilms only at 30C or below. We have still not analyzed the biofilm readout, but will do in the future.

How do the authors define the "accessory genes/genome". Are these simply all genes that are not part of the core genome? Could the authors comment on the completeness of the genomic sequences in the data set?

As suggested by the reviewer, we indeed consider those genes that are not present in the “core genome” (shared by all strains) as part of the so-called “accessory genome”. We are in fact following the definitions set up in the seminal pangenomics paper (Medini et al., 2005).

Regarding the completeness of the genome sequences, we believe that this is a fair point of discussion whenever draft genomes are involved. However, we are confident that the draft genomes we have collected/generated as part of this manuscript are of sufficient quality for nearly all genes to be correctly annotated. As a way of example we have compared the pangenome size rarefaction curves (computed using Roary, Page et al., 2015) between a cohort of 376 complete *E. coli* genomes vs. the genomes from our collection. In short, we have downloaded the complete genomes from NCBI’s refseq using the ncbi-genome-download script and annotated using prokka. We have then computed their pangeome using roary, using the same options that were used for the strains presented in our study. We have then plotted the pangenome size rarefaction curves, based on 10 shufflings of the strain’s order (Author response image 3).

**Author response image 3. respfig3:** 

The rarefaction curves are indeed very similar to each other; as expected the complete genomes set has a slightly higher number of total genes. This is expected given the completeness of these genomes. The difference with the strains presented in our study is however negligible. Furthermore, the core genome size of the RefSeq strains is composed of 2889 genes, versus 2815 for the genomes presented here, when considering 376 genomes. We have updated the methods section to report on the completeness of the presented genomes.

Were SNPs only called for sequences in common with the reference E. coli K12 genome? Would that potentially explain some of the discrepancies between the model predictions and the experimental outcome?

As the conditionally essential genes are derived from the reference strain (*E. coli* K12), we decided to call genetic variants with respect to this strain. As we have addressed in the discussion, we also believe that taking into account the impact of non-synonymous variants and the presence/absence of genes not present in the reference strain could improve our predictions. However, at the moment we cannot reliably assign a function to accessory genes in order to extend the predictive models in this way.

Does the term "mechanistic" appropriately describe the method used? Especially when one considers disruption scores in many areas of a genome and the conditional essentiality of genes?

We decided to adopt the term “mechanistic” as a contrast to more classical GWAS-like studies, which are based on statistical inference. Since the impact of nonsynonymous variants can be related to a protein’s functional constraints (i.e. SIFT) or structural stability (i.e. FoldX), we believe that the term “mechanistic” is appropriate. We have however added a short sentence in the Introduction explaining what we intend when using this term.